# Characterization of Anthocyanin Associated Purple Sweet Potato Starch and Peel-Based pH Indicator Films

**DOI:** 10.3390/foods10092005

**Published:** 2021-08-26

**Authors:** Mouluda Sohany, Intan Syafinaz Mohamed Amin Tawakkal, Siti Hajar Ariffin, Nor Nadiah Abdul Karim Shah, Yus Aniza Yusof

**Affiliations:** 1Department of Process and Food Engineering, Faculty of Engineering, University Putra Malaysia, Serdang 43400, Malaysia; mouludasohany@hstu.ac.bd (M.S.); hajarariffin@upm.edu.my (S.H.A.); nadiahkarim@upm.edu.my (N.N.A.K.S.); yus.aniza@upm.edu.my (Y.A.Y.); 2Department of Food Engineering and Technology, Faculty of Engineering, Hajee Mohammad Danesh Science and Technology University, Dinajpur 5200, Bangladesh; 3Laboratory of Halal Services, Halal Products Research Institute, Putra Infoport, Universiti Putra Malaysia, Serdang 43400, Malaysia

**Keywords:** purple sweet potato starch, anthocyanin, sweet potato peel, casting, film characterization, pH indicator film

## Abstract

In food packaging, smart indicator films based on natural resources have greatly attracted researchers to minimize the environmental issues as well as to satisfy consumer preferences for food safety. In this research, pH-sensitive films were prepared using purple-fleshed sweet potato starch (SPS) and sweet potato peel (SPP). Two categories of the film (i) SPS and (ii) SPS/SPP, were fabricated via solvent casting technique, incorporating different concentrations of commercial purple sweet potato anthocyanin (CA) at 0%, 1%, 1.5%, and 2% (*w*/*v*) and the physicochemical, mechanical, thermal, and morphological properties of the films were investigated. The thickness, water solubility, and swelling degree of the films increased with the increment of CA, whereas there were no significant changes in the water content (WC) of the films. Water vapor permeability (WVP) was decreased for SPS films while statistically similar for SPS/SPP films. The addition of CA reduced the tensile strength (TS) and tensile modulus (TM) yet increased the elongation at break (EaB) of the films as compared to films without CA. The FTIR results confirmed the immobilization of anthocyanin into the film. In SEM images, roughness in the surfaces of the CA-associated films was observed. A reduction of thermal stability was found for the films with anthocyanin except for the SPS/SPP CA 2% film. Furthermore, the CA-associated films showed a remarkable color response when subjected to pH buffers (pH 1 to 12) and successfully monitored chicken freshness. The fastest color migration was observed in acidic conditions when the films were immersed into aqueous, acidic, low fat, and fatty food simulants. The findings of this work demonstrated that the developed pH indicator films have the potential to be implemented as smart packaging to monitor food freshness and quality for safe consumption.

## 1. Introduction

The production and consumption of non-biodegradable synthetic films are an extreme threat to the ecosystem. As a substitute for plastic, the use of biopolymers for food packaging can play a vital role in minimizing this problem. Biopolymers are considered an important raw material for packaging films because of their plentiful availability, cheap rate, and biodegradability [1,2]. Biopolymers can be natural or synthetic, of which the natural one is more eco-friendly compared to the synthetic one [3]. Among the natural polymers, starch is assessed potential due to its large accessibility and comparatively low price [4,5]. Decent film-forming ability, biocompatibility, and biodegradability features make this hydrocolloid biopolymer the most promising packaging material [6]. To prepare starch-based biopolymers various sources of starch such as corn, rice, wheat, potato, and cassava along with other starchy foods such as yams, peas, and lentils are used as raw materials [7].

Sweet potato (*Ipomea batatas*) is a tropical starchy tuber crop and abundantly grown in Malaysia. In Malaysia, sweet potato production increased from about 26,582 tonnes in 2011 to 41,245 tonnes in 2017 [8]. Despite being the second most widely grown root vegetable in Malaysia, sweet potatoes remain under-utilized for making biofilms [9]. This crop is composed of skin, cortex, cambium ring, and different flesh colors including orange, yellow, purple, and cream. Starch is a major component of this root [10] approximately 50–80% of its dry weight [11]. This high starch content in sweet potato provides its good film-forming ability to produce biodegradable food packaging films that can replace petroleum-based films and promote sustainability. However, similar to other starches, sweet potato starch (SPS) has unconvincing mechanical and barrier properties than the commercial plastics and required some modification [12]. The addition of different functional additives into SPS can improve different properties of the starch films by changing the microstructure [13]. In addition to that, the effect of specific additives may include enhancement in food quality and safety [13]. Regarding additives, it is suggested to use biodegradable additives for biodegradable packaging because during storage the additional ingredients such as modifiers, coupling agents, plasticizers, fillers, catalysts, dyes, and pigments can migrate from the film to food products [14].

Furthermore, sweet potato peel (SPP) is usually discarded as waste after processing the sweet potato. This waste requires additional efforts for handling and might constitute an environmental issue. The SPP can be used as a filler into the film matrices to improve the properties of the film, thus the waste issue could be minimized. Concerning the waste issue, many researchers utilized food by-products in film casting. For instance, Shukor et al. [15] fabricated under-utilized jackfruit waste-based active packaging films, where the jackfruit waste including peel and straw was incorporated with thymol and tapioca starch. Othman et al. [16] studied the effect of banana pseudostem waste on the starch-based film. Kargarzadeh et al. [17] reinforced rice husk fiber into starch biocomposite films and Yang et al. [18] prepared wheat straw and polylactic acid biocomposite. To the best of the authors’ knowledge, research on active SPS films is relatively scarce, while several starch films have been reported on tuberous plants including potato starch [19,20,21,22], tapioca starch [3,23,24], cassava starch [25,26,27,28] and yam [29,30]. In this research, the starch and peel of purple sweet potato were utilized as starch-based films with pH-sensitive indicator dye.

In addition, general functions of food packages for example protection, containment, communication, and convenience are not sufficient to know the accurate information of shelf life of food products. Since food product quality might be influenced by several factors including improper handling, and temperature; it cannot be purely judged by the expiry date mentioned on the package [31]. Therefore, in response to the rising concerns of consumers regarding accurate food quality information, intelligent packaging has been developing great attention [20]. Intelligent or active packaging is proficient in detecting, sensing, tracking, recording, and showing properties to provide valid information about the status of food and its surrounding environment to consumers [32]. Among three types of intelligent packaging: indicators, data carriers, and sensors [33]; colorimetric pH indicator is novel, and it can monitor pH change due to food deterioration and/or extrinsic environment change and displays an apparent color change [34]. Freshness for several perishable foods is indicated by pH level since it produces reliable color responses with the degree of food deterioration during storage [35]. For instance, during protein breakdown, organic amines are produced in meats which causes an increase of pH in the food package; pH indicators can sense the change in pH and visually present to consumers [36]. In this way, consumers can get information about food quality according to a visual color difference without opening the packaging.

Generally, pigments and dyes are used for pH sensing. However, synthetic pigments are toxic [37] and therefore natural dyes are used as replacements in biodegradable packing materials [34,38,39,40]. Among natural plant pigments, anthocyanins are very sensitive to different ranges of pH due to their exceptional structure and readily change their color with the variation of pH solutions [20,35]. Anthocyanins from purple sweet potato are more stable when exposed to light and temperature compared to other plant dyes [41]. Several research works have successfully implemented this pigment as a pH indicator in the films based on cellulose nanofibres; agar/potato starch; carboxymethyl-cellulose/starch and Gellan gum [20,42,43,44]. However, none of the attempts systematically focused on the comparison of the film properties between SPS and SPS/SPP films containing anthocyanin.

Therefore, in this work, purple sweet potato anthocyanin (CA) at different loadings was immobilized into the SPS and SPS/SPP film matrices, and the physicochemical, mechanical, thermal, and morphological characteristics of the films were investigated. In addition, the color response of the indicator films to different pH ranges was analyzed as well as the migration of anthocyanin from the film to different food simulants was observed. Finally, the indicator films were employed to trace the freshness of raw chicken.

## 2. Materials and Methods

### 2.1. Materials

The purple sweet potato was obtained from a local farm located in Semenyih, Selangor Darul Eshan, Malaysia. *Anggun* is a well-known variety of purple sweet potato in Malaysia and there are three types of *Anggun* viz. *Anggun* 1, *Anggun* 2, and *Anggun* 3 that have been released by the Malaysian Agriculture and Development Institute (MARDI) in 2017. In this study, films were prepared from *Anggun* 1 due to their high production, long elliptical shape, and high anthocyanin content make them superior to other purple sweet potatoes [45]. The yield of purple-fleshed sweet potato starch was 7–8% with an amylose content of 26.9%. The commercial anthocyanin (CA) was obtained from the Osenai brand, China. The total monomeric anthocyanin content in CA was determined as 5.29 mg/g by pH-differential assay [46]. Buffer solutions from pH 1 to pH 12, acetic acid, ethanol (96%), calcium chloride, magnesium nitrate, and glycerol were purchased from R&M Chemicals, Malaysia.

### 2.2. Extraction of Purple Sweet Potato Starch (SPS)

White starch from purple sweet potato was extracted following the method of Soison et al. [10] with some modifications. After cleaning with tap water, fresh sweet potatoes were peeled and cut into pieces with around 3 mm thickness. The cuts were then macerated adding 1:1.5 (*w*/*w*) distilled water in a blender (FBG Blendy 125, Faber, Kuala lumpur, Malaysia), at full speed (10,000 rpm). Muslin cloth was used to filter the residues. The starch filtrate was kept undisturbed in a chiller (6 ± 2 °C) to settle for 3–4 h. The supernatant was poured out and the starch layer at the bottom of the beaker was resuspended in distilled water. The mixture was taken in the centrifuge tubes and centrifuged (Hettich Zentrifugen Universal 320, Merck KGaA, Darmstadt, Germany) at 9000 rpm for 3 min. The supernatant was discarded and distilled water was added again to proceed with the centrifugation. This process was continued until white starch sediment was obtained. The sediment starch was dried overnight in a convection oven (Jeio Tech OF-02G/12G/22G, Lab Companion, Seoul, South Korea) at 50 °C. The dried starch was cooled to ambient temperature (28 ± 2 °C) and ground using a laboratory-scale grinder (HR-20B-AEC, AEC Machinery (M) Sdn Bhd, Puchong, Malaysia) into powder. The powder was sieved through 100-micrometer mesh sieve size (AS200, Retsch, Dusseldorf, Germany) and packaged in polyethylene bags, sealed, and stored in a chiller (LF817LD, ASECO, Wen Ho Industries Sdn Bhd, Shah Alam, Malaysia) at 4 °C.

### 2.3. Preparation of Sweet Potato Peel (SPP) Powder

The SPP was dried overnight in a convection oven (Jepo Tech OF-22FW, Korea) at 50 °C. The dried peels were cooled to ambient temperature and ground using the same laboratory-scale grinder (HR-20B-AEC, AEC Machinery (M) Sdn Bhd, Puchong, Malaysia) into powder. The powder was sieved through 100-micrometer mesh sieve size (AS200, Retsch, Dusseldorf, Germany) and packaged in polyethylene bags, sealed, and stored for further use.

### 2.4. Preparation of SPS and SPS/SPP Films

The SPP films with different loadings of anthocyanin (SPS CA 0%, SPS CA 1%, SPS CA 1.5%, and SPS CA 2%) were obtained by adding 4 g of starch powder to 100 mL of distilled water in a beaker and the mixture was heated and stirred using a hot plate (Favorit, Malaysia). After it reaches 40 °C, 25% of glycerol (*v*/*w*) was added to the mixture. The mixture was stirred and heated at around 85 °C until gelatinization occurs for about 20 to 30 min. When the solution was cooled to below 40 °C, different loadings of anthocyanin powder (0%, 1%, 1.5%, and 2% (*w*/*v*)) were added accordingly and mixed properly for 10–15 min using the magnetic stirrer. Then the solution was sonicated (S-450D Branson Digital Sonifier, India) for 10 min to remove air bubbles. After cooling down the formulation was poured (35 mL) into the petri dish with a diameter of 140 mm and left for drying on a flat table for 2 days at 26 ± 2 °C. The steps were repeated to prepare SPS/SPP films with anthocyanin (SPS/SPP CA 0%, SPS/SPP CA 1%, SPS/SPP CA 1.5%, and SPS/SPP CA 2%) where the SPS and SPP powder was added in the ratio of 6:4.

### 2.5. Color Change of CA with pH

The color response of CA with a wide range of pH buffers was measured using a color spectrophotometer (HunterLab, Ultrascan Pro, Hunter Associates Laboratory, Inc., Virginia, USA) by measuring the CIELAB (International Commission on Illumination) coordinates (*L*, *a*, and *b*). One g CA was diluted with 300 mL of distilled water and centrifuged for 3 min at a speed of 9000 rpm. A total of 3 mL of the supernatant was added with 30 mL of each buffer solution ranging from pH 1 to pH 12. The color changes in the different pH buffers (1 to 12) were recorded. The color difference (Δ*E*) was calculated using Equation (1) [31]:(1)$$\Delta E=\sqrt{(L*-\;L{)}^{2}+(a*\;-\;a{)}^{2}+(b*\;-\;b{)}^{2}}$$
where *L** (99.98), *a** (0.15), and *b** (−0.13) were the color values of the standard white plate. The color parameters were averaged based on three replicates.

### 2.6. Characterization of Films

#### 2.6.1. Thickness

The thickness of the film was measured at ten random points using a hand-held micrometer (Digital micrometer 293-821-30, Mitutoyo Corporation, Kawasaki, Japan) with an accuracy of 0.001 mm, and the average value was determined.

#### 2.6.2. Water Content, Water Solubility, and Swelling Degree

The water content, water solubility, and swelling degree of film samples were determined according to the methods of Jamróz et al. [47] with slight modification. Each type of film was cut and the initial weight (W1) was taken (precision 0.0001 g); then dried in an oven at 70 °C for 24 h. The weight of the dried films was recorded to determine the initial dry matter (W2). Afterward, each film sample was immersed into 30 mL distilled water under constant agitation at 100 rpm using a shaking incubator (WiseCube Wis-20, Daihan Scientific Co., Ltd., Wonju, South Korea) for 24 h at room temperature (25 ± 2 °C) and dried with filter paper and weighed (W3). Next, the samples were dried in an oven at 70 °C for 24 h to determine the undissolved final dry weight (W4). Duplicate measurements were taken for each film sample to calculate the average value of the parameters. Water content, water solubility, and swelling degree were calculated by the following equations, respectively:(2)$$\mathrm{Water}\;\mathrm{content}\;(\%)=\frac{W1\;-\;W2}{W1}\;\times \;100$$
(3)$$\mathrm{Solubility}\;(\%)=\frac{W2\;-\;W4}{W2}\;\times \;100$$
(4)$$\mathrm{Swelling}\;\mathrm{degree}\;(\%)=\frac{W3\;-\;W2}{W2}\;\times \;100$$

#### 2.6.3. Water Vapor Permeability (WVP)

The WVP was determined by gravimetric methods using the water vapor permeability cup (No. 318 water vapor permeability cup, Yasuda Seiki Seisakusho Ltd., Nishinomiya, Japan) according to JIS Z 0208 method [3]. A mixture of paraffin wax and bee wax (8:2) was prepared and melted using a magnetic stirring hotplate. The water permeability cups were filled with 10 g of anhydrous calcium chloride desiccant to achieve 0% RH. The films were cut into circular shapes (7 cm diameter) and sealed to the cup using the wax solution. The cup was then placed in a desiccator containing saturated magnesium nitrate solution to provide a constant RH of 51% at 25 °C. A digital temperature humidity meter (Proskit NT-312, Techno Tools & Equipment Sdn Bhd, Ampang Jaya, Malaysia) was used to monitor the relative humidity and temperature inside the desiccator. Changes in the weight of the cup were recorded every 1 h for 2 days and plotted to obtain a weight loss versus time graph. Duplicate measurements were performed for WVP calculation for each film sample. The water vapor transmission rate (WVTR) was calculated using the following equation:WVTR = (W/t)/A(5)
where W/t is the slope of the weight change (g) versus time (h) graph and A is the transmission area of the film (28 cm^2^). The water vapor permeability (WVP) was calculated using the following equation:(6)$$\mathrm{WVP}=\frac{\mathrm{WVTR}\;\times \;L}{P1\;-\;P2}$$
where L is the average thickness of the film (mm), P1 is the water vapor partial pressure in the desiccator at RH of 51% (21.64 × 10^5^ Pa) and P2 is the water vapor partial pressure in the cup at RH of 0% (0 Pa).

#### 2.6.4. Colorimetric Analysis

The CIELAB coordinates (*L*, *a*, and *b*) were measured to determine the color of the prepared films. The color was recorded using a color spectrophotometer (HunterLab, Ultrascan Pro, USA) at five different places of each film. The total color difference and whiteness index of the films were then calculated. To compare the color change with pH, the anthocyanin added films were cut in a rectangular shape (2 cm × 1.5 cm) and immersed in 5 mL of each of the pH 1 to 12 buffers. After 10 min the films were removed, kept for drying for around 3–4 h, and the color was recorded with a 4 mm lens using a portable colorimeter (Precise color reader WR 18, Shenzhen Wave Optoelectronics Technology Co., Ltd., Shenzhen, China). The color difference (Δ*E*) and whiteness index was calculated using the following Equations [28,31]:(7)$$\mathrm{WI}=100-\sqrt{(100\;-\;L{)}^{2}+a^{2}+b^{2}}$$
where *L** = 87.89, *a** = 0.51, and *b** = 1.74 were the color values of the standard white plate of the portable colorimeter, and for the color spectrophotometer the values were *L** = 99.6, *a** = −0.11, and *b** = −0.22; meanwhile, *L*, *a*, *b* were the color values of the film samples.

#### 2.6.5. Mechanical Properties

Mechanical analysis was performed using a texture analyzer (TA.XT2 texture analyser, Stable Micro Systems, Godalming, UK) following ASTM D882 standard [48]. Film samples were cut into rectangular strips (100 mm × 15 mm) and placed between the grips. Initial grip separation and test speed were set to 50 mm and 0.5 mm s^−1^, respectively. Force and distance were recorded during the extension of the films to break. A minimum of three samples was tested for each type of film.

#### 2.6.6. Fourier Transform Infrared (FTIR) Spectroscopy

FTIR spectra using total attenuated reflectance (ATR) were obtained in Nicolet 6700-Thermo Nicolet, Thermo Fisher Scientific, Waltham, MA, USA. FTIR spectra were recorded using the frequency range of 650–4000 cm^−1^ at a resolution of 4 cm^−1^.

#### 2.6.7. Scanning Electron Microscopy (SEM)

The micrographs of the films were recorded by a JEOL model microscope (JEOL JSM 6400, JEOL Ltd., Tokyo, Japan) at an accelerating voltage of 10 kV. Samples were attached to double-sided adhesive tape and mounted on the specimen holder. Samples were sputtered and coated with a gold layer under vacuum (1 × 10^−4^ Torr) to increase their electrical conductivity.

#### 2.6.8. Thermogravimetric Analysis (TGA)

The thermal degradation profile was analyzed using TGA Instrument, Mettler Toledo (Model TGA/DSC 1, Mettler-Toledo AG Analytical, city, Schwerzenbach, Switzerland) following ASTM E1131 standard [49]. An amount of 5 mg to 15 mg film sample was placed into aluminum pans under nitrogen flow at a rate of 25 mL/min, in a temperature range from 25 °C to 500 °C with a heating rate of 10 °C/min.

### 2.7. Feasibility of Anthocyanin Migration

Food simulants with similar physicochemical properties to foods that mimic their behavior can be used to assay the migration of the additives [50,51]. To investigate the release of anthocyanin from films to aqueous food products (high water content food), distilled water (pH 5.8) was selected as a food simulant [50,52]. Correspondingly, three other food simulant solutions— (i) 3% acetic acid (resemblance to acidic food—pH 2.9); (ii) 50% ethanol (resemblance to lower fatty food—pH 5.3); and (iii) 95% ethanol (resemblance to fatty food—pH 6.4) were considered for the migration test [53]. The SPS and SPS/SPP films containing anthocyanin were cut into a square shape (3 cm × 3 cm) and immersed into 20 mL of each food simulant at room temperature (28 ± 2 °C). After 10 min, the color change was monitored and the fastest color migrated food simulant was determined.

### 2.8. Direct Food Contact

This test was conducted to evaluate the performance of the films as pH indicators at room temperature on the chicken meat sample. Room temperature (28 ± 2 °C) was considered to trace the spoilage of chicken meat using the indicator films. The chicken meat was cut into a rectangular shape and kept in Petri dishes. The SPS and SPS/SPP films containing anthocyanin were cut into a rectangular shape (3 cm × 1.5 cm) and placed on top of the samples. The samples were monitored at 0, 4, 16, and 24 h of the test period. The color change of the films was physically observed and the pH change of the chicken was examined using pH paper (Merck/USA pH Paper Universal Indicator Strips).

### 2.9. Statistical Analysis

The experimental data were analyzed with analysis of variance (ANOVA), and comparison between means was assessed by Fisher’s method using a 5% significance level. The analysis was performed using Minitab 16 (State College, PA, USA).

## 3. Results and Discussion

### 3.1. Physicochemical Properties of Film

#### 3.1.1. Fourier Transform Infrared Spectroscopy (FTIR)

The FTIR was performed to analyze the chemical interactions on the molecular structure of films after adding anthocyanin. Commercial anthocyanin (CA), SPS CA 0%, SPS CA 2%, SPS/SPP CA 0%, and SPS/SPP CA 2% were considered for FTIR operation, and the results are depicted in Figure 1.

All films exhibited a wide band around 3200–3500 cm^−1^, which were ascribed to the strong stretching vibration of a massive hydroxyl group (O–H) present in starch, glycerol, water [20,43,54,55], and sweet potato anthocyanin [42]. Nevertheless, a significant shift of O–H absorptions in the SPS CA 2% (3342 cm^−1^) and SPS/SPP CA 2% (3347 cm^−1^) film was found, which demonstrated new hydrogen bonds were formed between anthocyanin and starch that condensed the hydroxyl group interactions in the starch polymer [43]. The absorption peak between 2927–2934 cm^−1^ corresponded to the stretching vibration of the C–H methyl group [56]. Meanwhile, in the spectra of CA, a strong absorption peak at 1639 cm^−1^ and a characteristic peak at 1550 cm^−1^ were attributed to the bending vibration of C=C aromatic rings. A weak characteristic peak at 1262 cm^−1^ was assigned to the stretching of the pyran ring in flavonoid compounds, and an absorption band at 1026 cm^−1^ was attributed to C–H deformations of aromatic rings [35,42]. For the SPS CA 0% and SPS/SPP CA 0% films spectra, the band 1647 cm^−1^ and 1648 cm^−1^ was ascribed to the presence of tightly bound water [20,42]. In the spectrums, the band between 1417–1458 cm^−1^ and 1150–1152 cm^−1^ could be associated with the stretching of C–H and O–H groups respectively [31]. Several other absorption bands indicated the contribution of different functional groups such as group C–O (H) in the band between 1016–1031 cm^−1^ [54]. However, any noticeable band shifts in the FTIR spectra with the addition of any component means there is an interaction present between the components. For the SPS CA 2% and SPS/SPP CA 2% films spectra, the absorption became more intense (1645 cm^−1^ and 1644 cm^−1^) than the SPS CA 0% (1647 cm^−1^) and SPS/SPP CA 0% (1648 cm^−1^) films. This band shifting could be ascribed as the presence of aromatic ring stretches due to the interactions between anthocyanin and the polymer matrix. Thus, it can be concluded that anthocyanin was immobilized in the film matrix and formed strong physical and chemical interactions with the polymers [20,31,42].

#### 3.1.2. Surface Morphology

The SEM was used to investigate the surface morphology of the SPS and SPS/SPP films with and without anthocyanin and the images are presented in Figure 2. The SPS CA 0% film had a smooth surface and this might be due to the homogenous dispersion of glycerol in the film matrix. Moreover, the rough and protruding surface image of the SPS/SPP CA 0% film might be due to the SPP particles. The peel possibly caused heterogeneity in the film components and created a superficial area which resulted in the surface roughness.

When the anthocyanin was added, a slight rough structure was observed for the SPS CA 2% film. Luchese et al. [57] reported the surface roughness did not influence the color variation of the film as a function of pH. According to Prietto et al. [39], low interaction of the anthocyanin compound with starch and glycerol can cause roughness to the films. The SEM images implied that CA had admirable compatibility with SPS and SPS/SPP. The presence of glycerol plasticizer had perhaps improved the compatibility between the anthocyanin and starch as the hydroxyl groups of the plasticizers can form intermolecular hydrogen bonds with the hydroxyl groups of starch and CA. Thus, simultaneously reduced the intermolecular bonding and entanglements between polymer chains [40]. The interaction of CA with starch molecules had led to the formation of a continuous polymeric matrix which verifies the FTIR results.

#### 3.1.3. Thickness

Film thickness is an important parameter and it contributes to the barrier and mechanical properties of the film. In general, the SPS/SPP film was thicker than the SPS film and the thickness for both films increased with the addition of an increasing amount of anthocyanin (Table 1). This finding was in agreement with the report by K. Zhang et al. [31] who prepared pH-sensitive films based on starch/polyvinyl alcohol incorporating purple sweet potato anthocyanins at two different loadings. They found that the increment of the film thickness was from 64 µm to 71.6 µm for 0.5% and 97.7 µm for 1% anthocyanin incorporation. Prietto et al. [39] reported the thickness of the films is directly proportional to the concentration of solids in the formulation. Thus, the presence of glycerol and CA could be associated with the increase in thickness which was consistent with the results of other research [28,39,42,54,58]. Glycerol and anthocyanin could have induced the plasticizing effect by disrupting and restructuring intermolecular polymer chain networks, creating more free volumes that lead to thicker film [59]. According to Qin et al. [58] and Jiang et al. [42], the elevating amount of anthocyanin could improve the complexity of the film-forming matrices which can increase the film thickness.

#### 3.1.4. Water Content (WC)

In Table 1, there were no significant differences (*p* < 0.05) in the WC values of the SPS and SPS/SPP films. It has been reported that natural dyes for instance curcumin and purple sweet potato anthocyanin have no significant influence on the WC of starch/polyvinyl alcohol/glycerol films [34]. Researchers also mentioned that glycerol can affect the moisture content of the film as its hydrophilic nature can intersperse among and between polymer chains, disrupt hydrogen bonding, and spread the chains apart [34,60]. As the percent plasticizer added was fixed for all formulations in this research, therefore the results ascribed towards no significant influence of CA on the prepared films. However, Table 1 showed that the WC of SPS CA 0% (8.14%) film was slightly higher than the SPS/SPP 0% (7.50%) film. These results might be attributed to the good interactions of the SPS and SPP in the film structure. Along with SPS, SPP also contains starch and this increasing amount of starch could form more hydrogen bonds among the film components. The formation of new hydrogen bonds could restrict the free water molecules to interact as strongly with SPS/SPP CA 0% films compared with the SPS CA 0% films [61]. For the SPS films, the SPS CA 1% film contained less WC compared to SPS CA 0% film, and the WC values increased with the inclusion of a higher amount of CA (7.90%, 8.20%, and 8.37% for CA 1%, 1.5% and 2% correspondingly) in the SPS CA films. Xue Mei et al. [60] experienced no significant differences in the WC for sago starch films with anthocyanin from torch ginger extract and the WC values slightly increased with the increasing amount of anthocyanin. Regarding the SPS/SPP films, Table 1 exhibited that the WC of the SPS/SPP CA 1% film was higher than the SPS/SPP CA 0% film, and with the increasing amount of CA, the water content decreased (8.43%, 7.44% and 6.77% for CA 1%, 1.5%, and 2% respectively). Jiang et al. [42] also experienced a decrease in moisture content with the increasing amount of purple sweet potato anthocyanin in indicator films from carboxymethyl-cellulose. Zhai et al. [40] found a similar effect of roselle anthocyanin on the films with starch/polyvinyl alcohol. The WC trend for the SPS/SPP films was opposite to the SPS films which can be explained as the consequence of the presence of starch in the peel powder and the interaction among the increasing starch and other components of the film formulation.

#### 3.1.5. Water Solubility

Water solubility is an important parameter for intelligent film because the biobased films are attracted to water. Water solubility is the indicator for water resistance and barrier properties of the manufactured films. The use of film relies on film water solubility. Higher water solubility is not desired for packaging due to its limitation on aqueous food [31]. For example, insoluble films can be used on high-moisture foods while water-soluble films should be applied as a coating of fruits and vegetables, due to their facility to be removed by washing [62]. In this study, the water solubility of the SPS/SPP CA 0% film was higher than the SPS CA 0% film (see Table 1). The strong intermolecular force of starch molecules in the SPS films became weaker in the SPS/SPP films due to the presence of fiber from the peel as fiber can generate a weak film structure [63]. A higher value of water solubility was found for the SPS films with an increasing amount of CA. This increment might be the consequence of the water-soluble behavior of CA. Anthocyanin might have indirectly increased the hydroxyl group, thereby improved the affinity towards the water. These results were aligned with pH-sensitive films based on sago starch incorporated with anthocyanin from torch ginger and starch/polyvinyl alcohol associated with purple sweet potato or red cabbage anthocyanin film where a significant (*p* < 0.05) increment in the water solubility was observed with the enhancement of anthocyanin content [31,60]. Higher water solubility was also observed in films based on cassava starch when blueberry residue was added [26] and in corn starch films associated with black bean seed [39]. Prietto et al. [39] and Zhang et al. [31] mentioned the higher water solubility is attributable due to week intermolecular bonds in starch and the increase in the number of strong hydrophilic groups available for water absorption after interactions between the starch and anthocyanins.

#### 3.1.6. Swelling Degree

The swelling degree (Table 1) was significantly higher in the anthocyanin indicator films than the control SPS and SPS/SPP films. The SPS/SPP films showed a higher swelling degree than the SPS films which could be due to the presence of fiber in the peel as Luchese et al. [64] mentioned that presence of fiber can cause swelling. In addition, the peel particles might create a superficial area, surface pores, and channels, which can enhance the water uptake and swelling ability of film [65]. For both types of film, the SPS CA 0% and SPS/SPP CA 0% films demonstrated the lowest swelling degree (6.61% and 70.81% respectively) while the highest swelling degree was found for the SPS CA 2% and SPS/SPP CA 2% films (148.07% and 223.32% respectively). The enhancement in the swelling degree with *Clitoria Ternatea* flower (CTE) anthocyanin incorporation (from 51.75% to 76.69%) was also experienced by Koshy et al. [66] for intelligent starch-based biopolymer film. Luchese, Garrido, et al. [64] found the swelling percentage around 250% in the cassava starch films incorporated with blueberry pomace. An increment in the hydrophilic group can result in higher swelling in the films [13]. According to Halász and Csóka [67], the higher swelling degree is advantageous as aqueous media which changed pH can penetrate the bulk matrix and increase the pH sensitivity.

#### 3.1.7. Water Vapor Permeability (WVP)

WVP is an important parameter to determine the barrier properties of polymeric films. A low WVP value indicates a high sealing effect and ensures the preservation of food products. In Table 1, the WVP value was significantly higher for the SPS/SPP films than the SPS films. These differences were likely to occur because the presence of fiber in the peel interferes with the film matrix. Fiber can develop weak points in the film structure that promote changes in the WVP of the samples [63]. Versino and García [68] reported that the presence of filler can increase the tortuosity of the pathway for water molecules facilitating the water molecules to transport, thus, the WVP increases. In this study, the SEM images (Figure 2) showed rough and uneven structure for the SPS/SPP films compared to the SPP films which were in alignment with the WVP results. Regarding the SPS films, the WVP decreased when the CA was added and the values were 1.81, 1.34, 0.68, and 0.76 (g/Pa h m) × 10^−11^ for the SPS film with 0%, 1%, 1.5%, and 2% CA loadings, respectively (Table 2). The improved WVP result verifies this inner structure of the SPS CA 2% film (Figure 2). Xue Mei et al. [60] also found lower WVP values for the sago starch film with the incorporation of torch ginger extract (TGE). They obtained the WVP values of 3.3, 3.2, 2.9 and 2.6 (g m/day kPa m^2^) × 10^−4^ for 0%, 10%, 20% and 30% (*w*/*w*) TGE inclusion respectively. On the contrary, among the SPS/SPP films, there were no significant changes (*p* < 0.05) in the WVP with the addition of CA. The SEM result also indicated the WVP of SPS/SPP CA 2% was not affected by the structure of the film although the coarseness was increased for SPS/SPP CA 2% film. A similar observation was reported by Andretta et al. [26] whereby the WVP was the same (0.002 g mm h^−1^ m^−2^ kPa^−1^) as the cassava starch films after adding the blueberry residue. Several authors mentioned that the variations in WVP could be associated with many factors such as hydrophilic/hydrophobic ratio of the film components, film crystallinity, tortuosity effects, the presence of pores/voids, and structural defects [26,68,69]. According to Qin et al. [58] and Yun et al. [28], the new hydrogen bonding could reduce the availability of hydrophilic hydroxyl groups in the films, thereby improve the WVP. Andretta et al. [26] mentioned that the interaction between the water molecule and film matrix can make water desorption more difficult. From the morphology of the films, it can be postulated that the plasticizer, starch, and CA interactions resulted in enough compatibility in the films to improve the WVP.

### 3.2. Mechanical Properties of Films

The strength, durability, and resistance to extraneous force for being damage depends on the mechanical properties of packaging material. Tensile strength (TS) is the maximum force pull to the sample which indicates the ability to accept load or tension without damaging the composite film. Elongation at break (EaB) indicates the maximum change in film length while tensile strength is applied. Tensile modulus (TM) or elastic modulus is one of the basic measurements for testing the film rigidity. Figure 3 illustrates the tensile strength (a), elongation at break (b), and tensile modulus (c) values of the SPS and SPS/SPP films with and without CA. The TS and TM values of the SPS CA 0% film (TS-45.27 MPa and TM-1799.3 MPa) was statistically higher (*p* < 0.05) than the SPS/SPP CA 0% (TS-10.82 MPa and TM-317.8 MPa), whereas the EaB (8.09%) was lower compared to the SPS/SPP CA 0% film (EaB-20.73%). The possible explanation for this observation could be the presence of SPP particles that caused heterogeneity in the components of the film formulations which was verified in the SEM images (Figure 2), thus reduced the TS and TM. Cassava starch films containing higher particle-sized blueberry residue powder also unveiled a decrease in tensile strength and elastic modulus values [50]. According to Luchese et al. [50], the cohesion of the film materials containing heterogeneous particles affects the mechanical characteristics of the film if the particles are not completely compatible with the biopolymer. They also reported fiber addition encourages the increment of elastic modulus and a decrease in the elongation percentage. The consequence of these results indicated the particle size was dominant over fiber content and the uneven film structure promoted disruptions and points of discontinuity. The anthocyanin incorporation had significant effects on TS, EaB, and TM of the prepared films (Figure 3). Based on Figure 3a, the TS values of the CA-associated films for both types were remarkably less than the films without CA. Statistically, no changes in the TS of the SPS films with 1% and 2% anthocyanin whereas the SPS CA 1.5% possessed the highest value.

The TS values obtained for the SPS CA 1%, SPS CA 1.5%, and SPS CA 2% films were 1.03, 4.3 and, 0.86 MPa, respectively, indicated that certain loading of anthocyanin enabled TS enhancement while excessive anthocyanin had a negative impact. Piñeros-Hernandez et al. [69] experienced a similar result showing higher tensile strength for 10% rosemary extract compared to 5% and 20% (ranging from 0.5 to 0.8 MPa) when added to cassava starch films. Zhang et al. [31] mentioned that excessive anthocyanin loading can destroy the polymer network resulting in a decrease value of TS and EaB. For the SPS/SPP films with CA, there were no significant changes in the TS values although a slight decrease was observed with the higher CA content. On the contrary, the EaB percentage radically increased for the CA added films compared to the films without anthocyanin and the value decreased with the increasing amount of anthocyanin for both cases (Figure 3b). The SPS CA 1% film attained the highest EaB value (66.60%) which was statistically similar to the value of SPS CA 1.5% (63.18%) film. The SPA CA 2% obtained the lowest EaB value (56.57%) among the SPS CA films. A similar attribute was found in the SPS/SPP CA films, where the SPS/SPP CA 1% film contained the highest EaB value (34.52%); not statistically different from the SPS/SPP CA 1.5% film (34.36%) and the lowest value was possessed by the SPS/SPP CA 2% (23.04%) film. Figure 3c showed, among the SPS CA films, the highest TM value was recorded for the SPS CA 1.5% film (20.1 MPa). For the SPS/SPP CA films, although the values were statistically similar, 2% loading of anthocyanin resulted in the highest TM value (6.9 MPa).

Research on sago starch films with different loading of anthocyanin (0, 10, 20, 30% *w*/*w*) from torch ginger extract [60] and starch/polyvinyl alcohol (SPVA) films incorporated with different percentages (0, 30, 60, 120 mg/100 g) of roselle anthocyanins [40] experienced the tensile strength decreased (ranging from 5.00 to 4.26 N/m^2^ and from 48.97 to 41.85 MPa, respectively) and EaB increased (ranging from 48.55 to 85.14% and from 44.15 to 88.28%, accordingly) with the higher anthocyanin percentages. The TM decreased in the range from 82.48 to 73.96 MPa in the study conducted by Xue Mei et al. [60]. Prietto et al. [39] also observed reduced TS and increased EaB for pH-sensitive films based on corn starch with anthocyanins extracted from the black bean seed coat. Andretta et al. [26] reported cassava starch-based pH indicator films with blueberry residue possessed lower TS and TM and higher elongation values. The discussion specifies that composite material can be affected by the type and amount of blending compounds added. The tensile strength can be decreased with the addition of anthocyanin and because of the change in intermolecular forces [43]. According to Prietto et al. [39], the presence of anthocyanins can weaken the intermolecular interactions and thus affect the mechanical properties of the films. They also mentioned the plasticizing effect of water and anthocyanin can contribute to low TS and high EaB as well as low TM.

In this study, TS values were inversely proportional to water solubility similar to Prietto et al. [39] and Xue Mei et al. [60]. Generally, the EaB follows the opposite trend and TM follows a similar trend to tensile strength and this study observed likewise. An increase in the plasticity of the film and the inner structure of anthocyanin-associated films especially the films with higher CA concentration could increase the mobility and reduce the rigidity of the films. This study revealed anthocyanin incorporation in the SPS and SPS/SPP films has improved the elasticity with considerable tensile strength and rigidity.

### 3.3. Thermal Analysis

Figure 4 displays the thermogravimetric analysis (TGA) and derivative thermogravimetric (DTG) profiles of the prepared films. TGA revealed an almost similar degradation/ decomposition profile following three main steps for all the films. The pattern observed for the SPS CA 0% film was similar to cassava starch film described by Piñeros-Hernandez et al. [69]. Based on Figure 4, for all the films, the first stage exposed around 5 to 8% weight reduction between 50 °C to 150 °C which was attributed to water loss from the samples. Luchese, Garrido, et al. [64] mentioned, for cassava starch with blueberry pomace, the small weight loss close to 100 °C can be attributed to the evaporation of moisture content of the samples whereby Qin et al. [70] reported the temperature range from 30 °C to 185 °C for films based on cassava starch and anthocyanins from *Lycium ruthenicum* Murr. In addition, the anthocyanin in the CA films supposedly degraded in the first stage. Anthocyanins are thermally unstable and the stability of anthocyanins is strongly influenced by temperature [71]. Cai et al. [72] found that the maximum rate of weight loss for blueberry anthocyanin occurred at approximately 130.36 °C. They explained, the major reason was that anthocyanin was degraded to phenolic acids and aldehydes by deglycosylation and ring-opening reactions. For the SPS CA 0% film, although a slight change in the second stage weight loss was observed [64], in all the films, the second decomposition stage was exhibited around 190 °C to 300 °C, attributing to the glycerol rich phase volatilization and degradation similar to Qin et al. [70] with the decomposition temperature of 185 °C to 260 °C. Lozano-Navarro et al. [73] stated the decomposition temperature of glycerol was 290 °C in the study on chitosan-starch films with anthocyanin from natural extracts. Further heating above 300 °C induced the main thermal degradation of the SPS and SPS/SPP films.

The third weight reduction stage occurred from 300 °C to 390 °C, which reflected the major mass loss in the TGA due to hydroxyl dehydration. Zhang et al. [31] mentioned that a sharp mass loss between 280 °C and 350 °C was caused by the starch chain decomposition. In addition, Jumaidin et al. [74] who worked on sugar palm starch, have reported that this stage was ascribed to the elimination of hydrogen groups, decomposition, and depolymerization of the starch carbon chains.

The DTG curve peaks indicate the maximum weight loss rate of the sample. The DTG curve (Figure 4) exhibited that the anthocyanin affected the maximum weight loss rate of the films and the rate was higher for the SPS and SPS/SPP films without CA than the films with CA. Table 2 represents the derivative thermogravimetric (DTG) peak temperature and percent weight loss of the films at the third stage. From Table 2, in the third stage of thermal decomposition, the percent weight loss values of the SPS films were greater than the SPS/SPP films and anthocyanin incorporation decreased the weight loss rate for both types of films. In addition, the DTG peak temperature of the SPS films was higher than the SPS/SPP films. For each type, the thermal stability shifted towards a lower value when anthocyanin was added except for the SPS/SPP CA 2% film. For the CA-associated films, the highest thermal stability was found for the SPS CA 1.5% (355.95 °C) and SPS/SPP CA 2% (344.13 °C) films. These findings could be attributed to the anthocyanin and polymer matrix interaction.

Low thermal stability for the prepared indicator films based on tara gum/cellulose/grape skins extract and chitosan/corn starch/red cabbage extract was reported by Ma et al. [75] and Silva-Pereira et al. [35], whereas Qin et al. [70] experienced no effect on the thermal stability of films when *Lycium ruthenicum* Murr anthocyanin was added into the cassava starch films. However, in this study, the stability of the films at temperatures below 100 °C supports the potentiality of the films to be used in food packaging.

### 3.4. Color Analysis

#### 3.4.1. Color Parameters of Prepared Films

As shown in Figure 5, the SPS films were brighter than the SPS/SPP films. The SPS CA 0% film was colorless whereas the SPS/SPP CA 0% film was yellowish due to the presence of peel. The naturally occurring anthocyanin in the peel might have degraded during the peel powder processing and film making that resulted in the film color being yellowish.

It has been reported that during thermal processing the anthocyanin can be directed to thermal degradation and form colorless or undesirable brown-colored polymeric pigments [76]. Due to the color of the SPS/SPP CA 0% films, visually, the SPS/SPP films with CA were darker compared to the SPS films with CA and burgundy in color. The color parameters of different films were summarized in Table 3. Significant differences (*p* < 0.05) were detected in all the color parameters (*L*, *a*, *b*, and Δ*E*) of the films. The *L* and WI value confirmed that the SPS/SPP CA films were darker than the SPS CA films and the value decreased with the increasing loading of anthocyanin.

Increasing *a* value means the redness of the films increased. For the SPS CA films, the higher *b* value with additional anthocyanin indicated color shifting towards yellowness, at the same time the SPS/SPP CA films exhibited the opposite trend indicating the higher loading of anthocyanin causing blueness to the films. In addition, the total color difference (Δ*E*) was significantly higher in the SPS/SPP films than in the SPS films which signified variations in appearances in terms of color. The whiteness index (WI) values significantly decreased with the increase of CA content for both types of films and the SPS CA films possessed higher values than the SPS/SPP CA films. A similar range for *L*, Δ*E*, and WI values was observed for cassava starch films with Chinese bayberry anthocyanin [28]. The results concluded that there were variations in colors and darkness with the additional loading of CA. The changes in the color parameters with different anthocyanin content were also observed by Qin et al. [70] for cassava starch films incorporated with *Lycium ruthenicum* Murr anthocyanin.

#### 3.4.2. Color Change of CA Solution

The color change of CA in different buffer solutions (pH 1–12) was analyzed and a noticeable variation in color response was observed which was effortlessly recognized by the naked eye (Figure 6). The anthocyanin revealed a reddish/pink color from pH 1 to 4 which changed from purple to blue at pH 5 to 9. Accordingly, the color transformed into green and yellow at pH 10 to 12.

The color of anthocyanidins differs with the number of hydroxyl groups, attached to their molecules [77]. Changes in pH can also cause reversible structural transformations in anthocyanins molecules, which has a dramatic effect on their color [78]. The pH sensitivity of anthocyanins comes from the ionic nature of anthocyanin. In acidic solutions, the structure of anthocyanins changed to flavylium cation (red) and quinoidal anhydrase (purple). In the alkaline solutions, the structure of anthocyanins changed to quinoidal (green) and chalcone (yellow) [79,80]. Table 4 showed the color parameters (*L*, *a*, *b*, and Δ*E*) obtained from the color variations of CA solutions at pH 1–12 solutions. The *L* value of 0–50 and 51–100 indicates darkness and lightness of color respectively. Positive *a* (+*a*) value signifies redness and negative *a* (−*a*) value greenness; whereas positive *b* (+*b*) value is the measure for yellowness and negative *b* (−*b*) for blueness. A decrease in values of parameter *a* from positive to negative indicated that the color of the anthocyanin solutions changed from red to green and the green color has greater intensity towards a higher pH range [55]. The change in the *b* value from negative to positive signified the shifting of color from blue to yellow. The higher *b* value specified deeper yellow color which is achievable at a higher pH range [31]. However, a change in *L* values could not be observed through the naked eye due to the different colors of anthocyanin at various pH ranges [43]. The total color difference Δ*E* indicated a visually perceptible change of color and the greater Δ*E* number represented good visual color variability [55].

In Table 4, significant changes can be observed in the color parameters (*L*, *a*, *b*, and Δ*E*) (*p* < 0.05) at pH 1–12; which illustrated that CA was sensitive to pH change with good visual color variability and can be used as a pH-sensitive dye.

#### 3.4.3. Color Change of Anthocyanin Incorporated SPS and SPS/SPP Films at pH 1–12

The color responses of the SPS and SPS/SPP films containing CA at a wide range of pH (1–12) were observed and the images are depicted in Table 5. The change in colors was visible for all the samples and the SPS/SPP CA films were darker than the SPS CA films. Based on Table 5, Table 6 and Table 7, the trend of film color response was similar to the CA (Section 3.4.2), where reddish/pink color was observed at pH 1 to 4, purple to blue at pH 5 to 9, and green and yellow color at pH 10 to 12. Correspondingly, the SPS and SPS/SPP films with 2% CA exhibited bolder color than 1% and 1.5% loading of CA due to the rich anthocyanin content. The recorded values of *L*, *a*, *b*, and Δ*E* in Table 6 and Table 7 differentiated the color responses of the films. The high *L* value signifies the lightness and the value was greater for the SPS CA films compared to the SPS/SPP CA films. For all films, *a* value was higher for the acidic buffers indicating the redness of the films, and the value reduced as the pH buffer number increased which means the colors shifted towards green. On the contrary, the increasing tendency of *b* value with the increase of pH range was attributed to the color shifting from blue towards yellow. The color alteration of the films was the result of the structural transformation of anthocyanin at different pH [31]. It was ascribed to the flavylum cation in acidic conditions turning to anionic quinoidal as the pH shifted to alkaline [81]. Total color difference (Δ*E*) values demonstrated profound visual differences and the value was higher for the SPS/SPP CA film than the SPS CA films. Significant changes (*p* < 0.05) were observed for all the values (*L*, *a*, *b*, Δ*E*) indicated that the anthocyanin films in this study possessed a noticeable color response towards different pH values and could be implemented as pH indicator films.

#### 3.4.4. Direct Food Contact

There is a close relationship between the pH and freshness of foods. For example, during the spoilage of chicken meat, the pH changes from 5.8 to 7.4 due to the breakdown of protein and production of amines whereby the pH indicator can sense these changes and respond in accordance [82].

Therefore, the prepared CA associated films in this study could be employed to monitor the freshness of chicken. The rectangular film cuts were placed on chicken pieces at room temperature (28 ± 2 °C). From Table 8, the color change of the films with the pH change of the chicken at 0 h, 4 h, 16 h, and 24 h can be seen.

The pH of the chicken meat was found around 5, 6, 7, and 8 at 0, 4, 16, and 24 h respectively. Although no visible change was found in chicken appearance, offensive off-odor was experienced with time indicating the spoilage of chicken. For the SPS CA 0% and SPS/SPP CA 0% films, there was no change in color of the film throughout the 24 h of storage time. However, changes in color for the SPS CA 2% and SPS/SPP CA 2% films were observed after 4 h of storage. Figure 7 shows the color variations of the SPS CA 2% and SPS/SPP CA 2% films as the pH changed in the chicken sample with time. The figure showed that after 4 h, the SPS CA 2% film became darker with the pH approximately at pH 6. Subsequently, at 16 h (around pH 7) and 24 h (near to pH 8), the color was changed into fade color with the increasing pH of the chicken sample. At the same time, the color of the SPS/SPP CA 2% became darker as the pH increased in the chicken sample. These color changes were similar to the color response of the films mentioned in Section 3.4.3.

Although CA-associated SPS and SPS/SPP films successfully indicated the freshness or spoilage of chicken as a pH indicator by responding to pH variations, the migration of the anthocyanin color to the chicken meat was also observed (see Figure 8). This could be due to the high hydrophilicity of the starch-based films and the water-soluble behavior of anthocyanin. The color migration of anthocyanin from the SPS/SPP CA 2% film was more intense than the SPS CA 2% film. The structure of the SPS/SPP CA 2% film (shown in SEM result), the hydrophilic behavior of starch and CA as well as the darker color of the film might result in this strong migration compared to the SPS CA 2% film. Luchese, Abdalla, et al. [50] also observed the color migration from film to chicken meat while testing the application of blueberry residue incorporated cassava starch-based pH indicator films. The hydrophilicity of the films can be reduced by incorporating hydrophobic components into the film matrix; thus, the color migration can be minimized. Modification of starch or starch nanoparticles, usage of compatibilizers, or preparation of multilayer films are some ways suggested by Flores et al. [83] to decrease the hydrophilicity.

To improve hydrophobicity, reduction of water sensitivity and surface modification could be other options [84]. However, since the SPS and SPS/SPP films associated with CA were able to trace the pH change of chicken during storage by changing their color, with some modifications these films can be implemented as pH indicator films to detect the freshness of the food products.

### 3.5. Anthocyanin Migration Feasibility Test

#### Migration from Film to Food Simulants

The direct contact of packaging films with food products desires to examine the possibility of anthocyanin release from the film to the food. Luchese, Abdalla, et al. [50] and Ribeiro-Santos et al. [51] mentioned that the migration depends on the film structure and the chemical nature of the migrants. To determine the release of CA from films, the immersed film samples in different food simulants (i) distilled water–resemblance to aqueous food with pH 5.8, (ii) 3% acetic acid–resemblance to acidic food with pH 2.9; (iii) 50% ethanol–resemblance to lower fatty food with pH 5.3; and (iv) 95% ethanol–resemblance to fatty food with pH 6.4; were monitored and the color change of the solution at 0, 5, and 10 min at room temperature (28 ± 2 °C) was observed.

As it is depicted in Figure 9, for both the SPS and SPS/SPP films, after 5 min, a slight color change in the distilled water and 50% ethanol was observed where the color was intense for the acetic acid solution. At 10 min, deeper color was found in the solutions with bold pink color for the acetic acid solution. In contrast, there was no visible color change for 95% ethanol. A similar color change for simulants was observed by Luchese, Garrido, et al. [64] for cassava starch films incorporated with blueberry pomace. Buonocore et al. [85] reported that many factors could stimulate the release of active compounds from bio-based films, including chemical composition, the structure of film matrix, interaction among active components, and surrounding media. Considering that, the color change in distilled water could be attributed to the fast water penetration into the film matrix, allowing the diffusion of anthocyanin. On the contrary, the results for ethanol could be explained as only a low amount of ethanol can penetrate the film matrix because starch matrices are less swellable in ethanolic solutions [69].

Furthermore, anthocyanins are pH sensitives and change colors when subjected to different pH buffers due to transformations into several structural forms [86]. Due to the sensitiveness of anthocyanin to acid, anthocyanin interacted faster, taking the shortest time to migrate to the acetic acid solution. The pink coloration of the acetic acid solution was due to anthocyanin changes its color to reddish/pink in acidic conditions (Section 3.4.2). The distinguished color change intensity of the SPS CA 2% and SPS/SPP CA 2% might be due to the original color of the films (Figure 5).

## 4. Conclusions

Purple sweet potato starch and peel films (SPS and SPS/SPP) incorporated with different concentrations of purple sweet potato commercial anthocyanin (CA) were developed successfully. After the addition of CA into the SPS and SPS/SPP films, the thickness, water solubility and swelling of the films increased whereas the moisture content was not significantly affected to both types of films. The WVP of the SPS films decreased while the WVP of the SPS/SPP films remain unchanged. The mechanical properties were also influenced after the addition of anthocyanin. The films presented their ability to visual color responses (pink to yellow/green) over a wide pH range (pH 1–12), in which the SPP CA films showed more color variability, although the color change was bold for the SPS/SPP CA films. Furthermore, the FTIR result indicated strong interaction between anthocyanin and film matrix. Morphological scanning depicted increased roughness in the surface after CA incorporation. Substantial thermal stability was revealed by the thermogravimetric analysis. The fastest color migration was observed in acidic conditions when the films were immersed into aqueous, acidic, low fat, and fatty food simulants. In addition, the application of the films as pH indicators was conducted for monitoring chicken freshness. The CA associated films displayed visual color changes responding to the pH changes of chicken during spoilage. The results of this study signify that the CA-associated SPS and SPS/SPP films have the potential to be used as pH indicator package to monitor the freshness or spoilage of food products to ensure their quality and safety.

## Figures and Tables

**Figure 1 foods-10-02005-f001:**
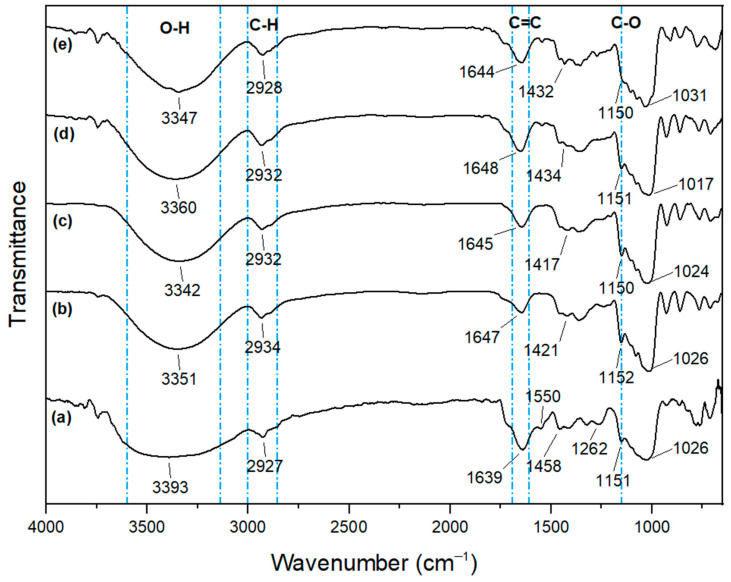
FTIR spectra of (**a**) CA (**b**) SPS CA 0% film, (**c**) SPS CA 2% film, (**d**) SPS/SPP CA 0% film, (**e**) SPS/SPP CA 2% film.

**Figure 2 foods-10-02005-f002:**
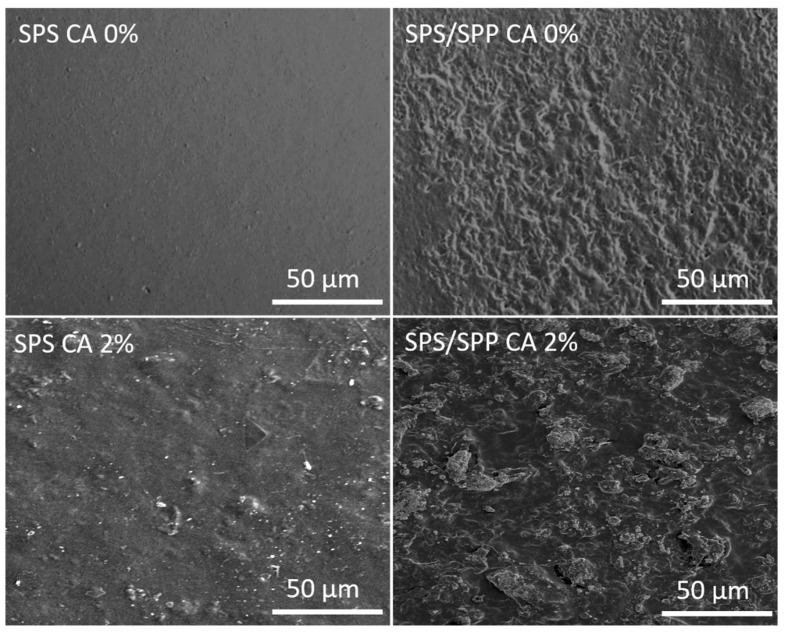
SEM micrographs of SPS and SPS/SPP films at CA of 0% and 2%.

**Figure 3 foods-10-02005-f003:**
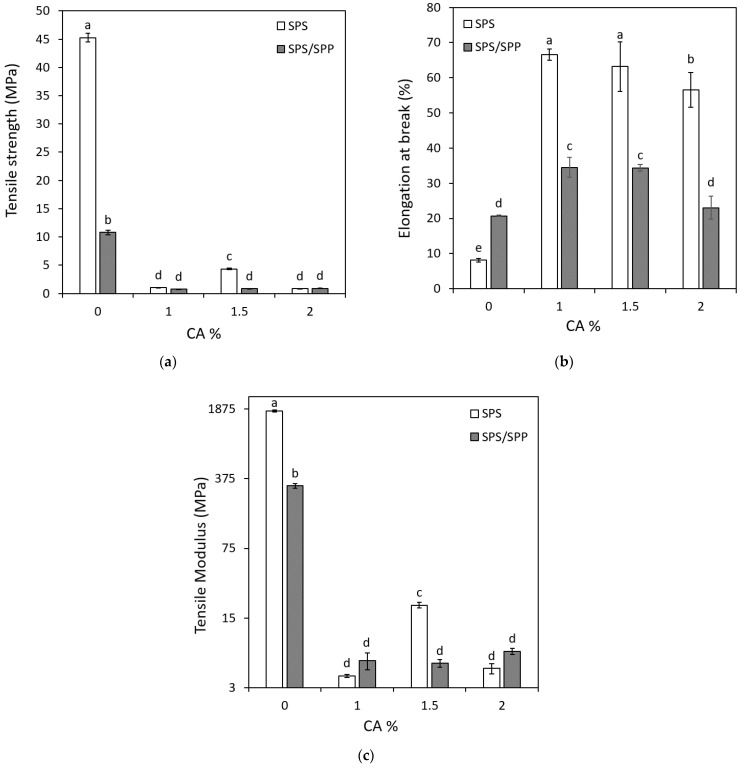
(**a**) TS, (**b**) EaB, and (**c**) TM of SPS and SPS/SPP films at CA of 0%, 1%, 1.5%, and 2%. The lowercase letters in the bars indicate significant differences by the Fisher’s test (*p* < 0.05).

**Figure 4 foods-10-02005-f004:**
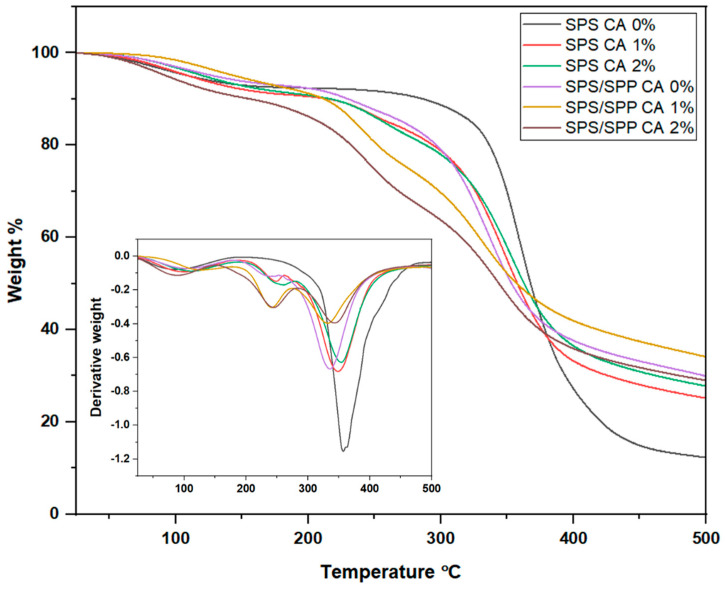
TGA and DTG thermographs of SPS and SPS/SPP films at CA of 0%, 1%, and 2%.

**Figure 5 foods-10-02005-f005:**
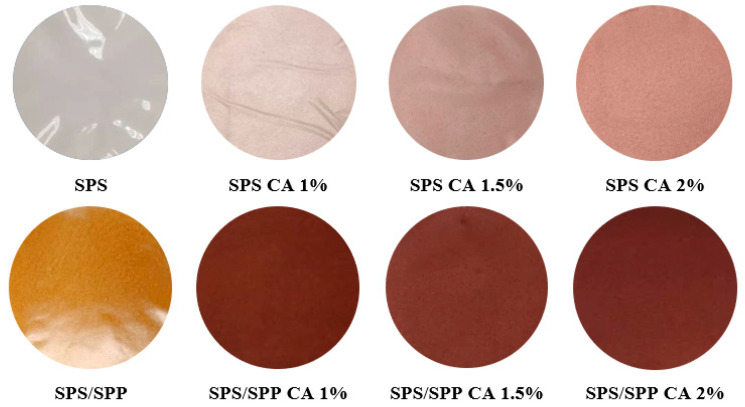
Visual appearances SPS and SPS/SPP films at CA of 0%, 1%, 1.5%, and 2%.

**Figure 6 foods-10-02005-f006:**
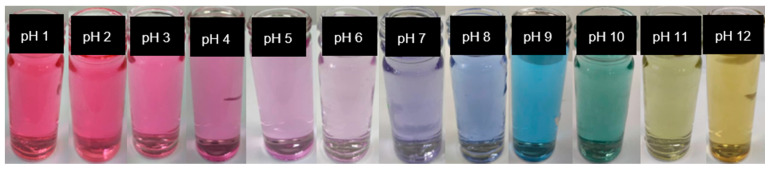
Color variations of CA in different buffer solutions (pH 1 to pH 12).

**Figure 7 foods-10-02005-f007:**
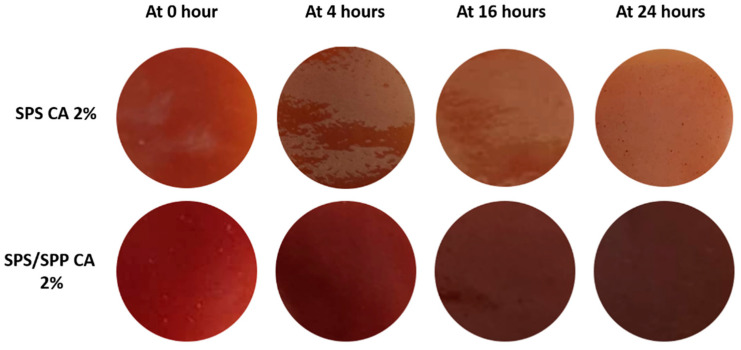
Color response of SPS CA 2% and SPS/SPP CA 2% indicator films with the change in pH of chicken deterioration with time at room temperature (28 ± 2 °C).

**Figure 8 foods-10-02005-f008:**
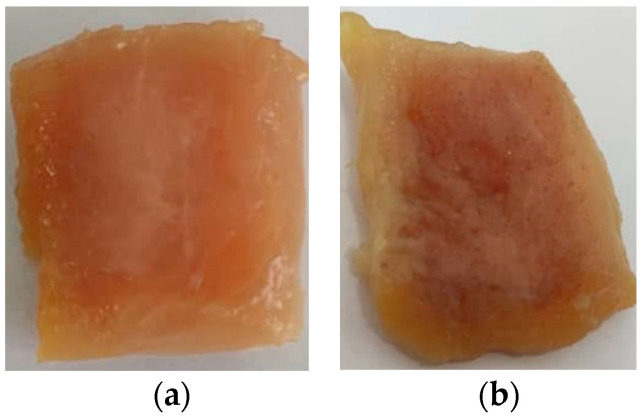
Migration of anthocyanin from (**a**) SPS CA 2% and (**b**) SPS/SPP CA 2% films into the chicken meat sample.

**Figure 9 foods-10-02005-f009:**
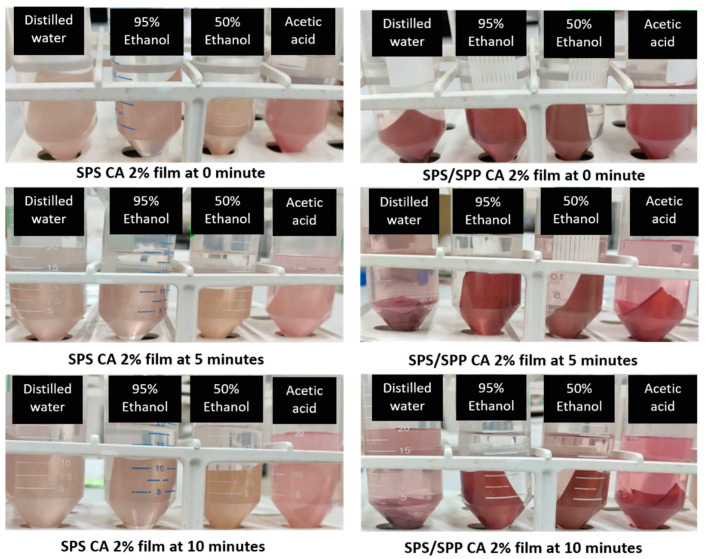
Anthocyanin migration from film to food simulants.

**Table 1 foods-10-02005-t001:** Thickness, water content, water-solubility, swelling, and water vapor permeability (WVP) of the SPS and SPS/SPP films at CA of 0%, 1%, 1.5%, and 2%.

Sample	Thickness (mm)	Water Content (%)	Water Solubility (%)	Swelling (%)	WVP (10^−11^ g/Pa h m)
SPS CA 0%	0.103 ± 0.001 ^g^	8.14 ± 0.34 ^a^	15.4 ± 0.19 ^c^	6.61 ± 0.55 ^h^	1.81 ± 0.12 ^ab^
SPS CA 1%	0.138 ± 0.004 ^f^	7.90 ± 0.67 ^a^	45.23 ± 10.29 ^b^	98.34 ± 0.23 ^e^	1.34 ± 0.35 ^ab^
SPS CA 1.5%	0.155 ± 0.005 ^e^	8.20 ± 0.47 ^a^	65.29 ± 0.81 ^a^	91.87 ± 0.19 ^f^	0.68 ± 0.08 ^b^
SPS CA 2%	0.205 ± 0.003 ^c^	8.37 ± 1.17 ^a^	60.72 ± 5.14 ^a^	148.07 ± 2.73 ^b^	0.76 ± 0.58 ^b^
SPS/SPP CA 0%	0.161 ± 0.002 ^d^	7.50 ±1.01 ^a^	22.83 ± 1.63 ^c^	70.81 ± 1.15 ^g^	2.36 ± 0.79 ^a^
SPS/SPP CA 1%	0.209 ± 0.006 ^c^	8.43 ± 0.82 ^a^	57.26 ± 2.88 ^ab^	128.68 ± 0.96 ^c^	2.45 ± 0.40 ^a^
SPS/SPP CA 1.5%	0.219 ± 0.004 ^b^	7.44 ± 0.74 ^a^	57.60 ± 5.93 ^ab^	102.94 ± 0.96 ^d^	2.51 ± 0.90 ^a^
SPS/SPP CA 2%	0.238 ± 0.006 ^a^	6.77 ± 0.16 ^a^	60.58 ± 7.24 ^a^	223.32 ± 2.38 ^a^	2.37 ± 0.85 ^a^

Different superscript lowercase letters within the same column indicate significant differences by the Fisher’s test (*p* < 0.05).

**Table 2 foods-10-02005-t002:** Derivative thermogravimetric (DTG) peak temperatures and percent weight loss of SPS and SPS/SPP films at CA of 0%, 1%, 1.5%, and 2%.

Samples	DTG Peak Temperature (°C)	Weight Loss (%)
SPS CA 0%	356.83	56.34
SPS CA 1%	350.54	43.45
SPS CA 1.5%	355.95	29.79
SPS CA 2%	354.71	31.81
SPS/SPP CA 0%	335.90	40.34
SPS/SPP CA 1%	332.20	30.04
SPS/SPP CA 1.5%	335.16	28.73
SPS/SPP CA 2%	344.13	26.27

**Table 3 foods-10-02005-t003:** Color parameters (*L*, *a*, *b*, Δ*E* and WI) of SPS and SPS/SPP films at CA of 0%, 1%, 1.5%, and 2%.

Sample	*L*	*a*	*b*	Δ*E*	WI
SPS	91.95 ± 0.06 ^a^	−0.92 ± 0.09 ^f^	6.19 ± 0.11 ^g^	10.01 ± 0.08 ^g^	89.80 ± 0.07 ^a^
SPS CA 1%	75.32 ± 0.83 ^b^	8.68 ± 0.44 ^e^	13.17 ± 0.49 ^e^	29.09 ± 1.03 ^f^	70.71 ± 1.03 ^b^
SPS CA 1.5%	63.87 ± 1.30 ^c^	16.55 ± 0.65 ^d^	14.33 ± 0.41 ^d^	42.03 ± 1.47 ^e^	57.75 ± 1.47 ^c^
SPS CA 2%	55.03 ± 0.45 ^e^	20.61 ± 0.66 ^c^	18.21 ± 0.64 ^c^	52.50 ± 0.77 ^d^	47.28 ± 0.76 ^d^
SPS/SPP	58.33 ± 0.87 ^d^	16.93 ± 0.61 ^d^	41.28 ± 0.45 ^a^	60.97 ± 0.51 ^c^	38.94 ± 0.51 ^e^
SPS/SPP CA 1%	37.56 ± 0.26 ^f^	22.88 ± 0.35 ^a^	19.09 ± 0.49 ^b^	68.93 ± 0.25 ^b^	30.81 ± 0.25 ^f^
SPS/SPP CA 1.5%	33.47 ± 0.42 ^g^	21.53 ± 0.58 ^b^	13.20 ± 1.03 ^e^	70.88 ± 0.39 ^a^	28.83 ± 0.38 ^g^
SPS/SPP CA 2%	32.12 ± 0.33 ^h^	20.89 ± 0.90 ^bc^	11.67 ± 0.92 ^f^	71.68 ± 0.50 ^a^	28.02 ± 0.49 ^g^

Different superscript lowercase letters within the same row indicate significant differences by the Fisher’s test (*p* < 0.05).

**Table 4 foods-10-02005-t004:** Color parameters (*L*, *a*, *b*, Δ*E*) of CA pH 1 to 12.

pH Value	*L*	*a*	*b*	Δ*E*
pH 1	55.98 ± 0.02 ^j^	75.69 ± 0.04 ^a^	−14.19 ± 0.07 ^i^	88.58 ± 0.02 ^a^
pH 2	62.91 ± 0.02 ^h^	65.22 ± 0.08 ^c^	−10.56 ± 0.10 ^h^	75.61 ± 0.07 ^c^
pH 3	56.69 ± 0.96 ^i^	73.81 ± 0.82 ^b^	−10.5 ± 0.57 ^h^	86.08 ± 0.29 ^b^
pH 4	62.95 ± 0.03 ^h^	52.9 ± 0.20 ^d^	−17.31 ± 0.05 ^j^	66.70 ± 0.19 ^d^
pH 5	69.65 ± 0.03 ^g^	26.07 ± 0.11 ^e^	−13.89 ± 0.07 ^i^	42.21 ± 0.10 ^g^
pH 6	73.52 ± 0.06 ^e^	12.93 ± 0.08 ^f^	−7.78 ± 0.07 ^g^	30.36 ± 0.10 ^i^
pH 7	74.29 ± 0.01 ^d^	−2.29 ± 0.04 ^g^	3.71 ± 0.03 ^e^	26.09 ± 0.01 ^k^
pH 8	74.35 ± 0.02 ^d^	−9.36 ± 0.07 ^k^	2.08 ± 0.10 ^f^	27.43 ± 0.02 ^j^
pH 9	70.18 ± 0.03 ^f^	−21.73 ± 0.03 ^l^	9.61 ± 0.10 ^d^	38.23 ± 0.04 ^h^
pH 10	78.24 ± 0.01 ^b^	−8.72 ± 0.05 ^j^	40.98 ± 0.06 ^c^	47.34 ± 0.06 ^f^
pH 11	77.00 ± 0.03 ^c^	−4.65 ± 0.03 ^h^	54.74 ± 0.06 ^b^	59.68 ± 0.06 ^e^
pH 12	84.32 ± 0.18 ^a^	−5.87 ± 0.06 ^i^	57.32 ± 0.15 ^a^	59.85 ± 0.18 ^e^

Different superscript lowercase letters within the same row indicate significant differences by the Fisher’s test (*p* < 0.05).

**Table 5 foods-10-02005-t005:** Color response images of the SPS and SPS/SPP films containing CA at pH 1 to 12.

pH Value	Images after the Color Change
SPS CA 1%	SPS CA 1.5%	SPS CA 2%	SPS/SPP CA 1%	SPS/SPP CA 1.5%	SPS/SPP CA 2%
pH 1	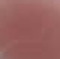	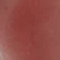	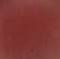	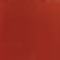	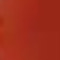	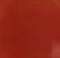
pH 2	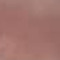	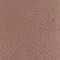	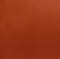	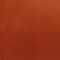	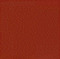	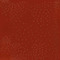
pH 3	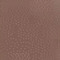	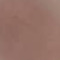	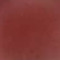	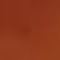	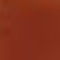	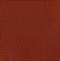
pH 4	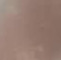	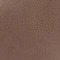	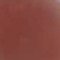	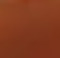	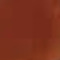	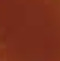
pH 5	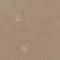	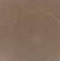	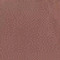	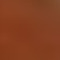	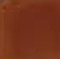	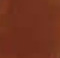
pH 6	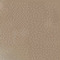	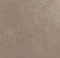	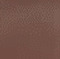	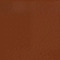	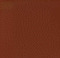	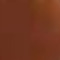
pH 7	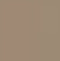	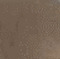	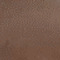	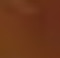	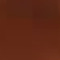	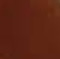
pH 8	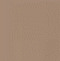	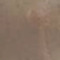	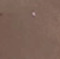	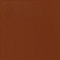	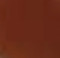	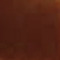
pH 9	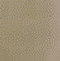	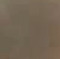	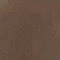	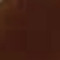	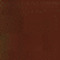	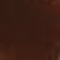
pH 10	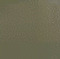	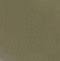	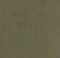	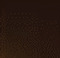	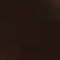	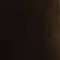
pH 11	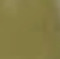	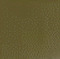	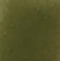	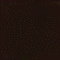	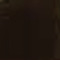	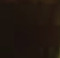
pH 12	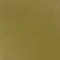	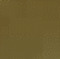	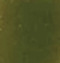	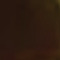	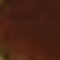	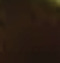

**Table 6 foods-10-02005-t006:** Color response values for the SPS CA 1%, SPS CA 1.5%, and SPS CA 2% films at pH 1–10.

**SPS CA1%**				
**pH Value**	** *L* **	** *a* **	** *b* **	**Δ*E***
pH 1	62.28 ± 0.67 ^ab^	18 ± 0.54 ^a^	10.08 ± 0.44 ^f^	32.12 ± 0.40 ^c^
pH 2	62.01 ± 1.77 ^ab^	13.31 ± 0.24 ^b^	10.67 ± 0.63 ^def^	30.24 ± 1.26 ^d^
pH 3	59.26 ± 3.34 ^cd^	12.06 ± 1.10 ^c^	10.31 ± 1.57 ^ef^	32.14 ± 2.33 ^c^
pH 4	61.14 ± 1.64 ^bc^	9.93 ± 0.39 ^d^	10.42 ± 0.70 ^ef^	29.68 ± 1.15 ^d^
pH 5	54.50 ± 1.86 ^e^	7.28 ± 0.98 ^e^	10.38 ± 2.07 ^ef^	32.24 ± 1.86 ^b^
pH 6	64.1 ± 1.27 ^a^	6.94 ± 0.40 ^e^	12.24 ± 0.75 ^cd^	26.80 ± 0.77 ^e^
pH 7	64.56 ± 0.35 ^a^	6.11 ± 0.07 ^f^	13.13 ± 0.20 ^c^	26.56 ± 0.24 ^e^
pH 8	62.01 ± 2.07 ^ab^	9.44 ± 0.28 ^d^	12.86 ± 0.33 ^c^	29.57 ± 1.67 ^d^
pH 9	58.76 ± 0.80 ^cd^	2.83 ± 0.08 ^g^	12 ± 0.42 ^cde^	30.97 ± 0.60 ^cd^
pH 10	59.03 ± 0.61 ^cd^	0.9 ± 0.07 ^i^	13.69 ± 0.33 ^c^	31.23 ± 0.49 ^cd^
pH 11	58.17 ± 1.27 ^d^	1.89 ± 0.14 ^h^	25.24 ± 1.82 ^b^	37.95 ± 0.28 ^a^
pH 12	60.96 ± 0.84 ^bc^	3.24 ± 0.07 ^g^	30.6 ± 0.52 ^a^	39.57 ± 0.16 ^a^
**SPS CA 1.5%**				
**pH Value**	** *L* **	** *a* **	** *b* **	**Δ*E***
pH 1	50.32 ± 1.46 ^c^	24.21 ± 0.95 ^a^	11.63 ± 0.74 ^cd^	45.53 ± 0.70 ^de^
pH 2	60.83 ± 1.36 ^a^	16.49 ± 0.86 ^b^	14.10 ± 0.61 ^a^	33.77 ± 1.70 ^h^
pH 3	57.90 ± 0.51 ^a^	14.19 ± 0.14 ^c^	13.41 ± 0.17 ^ab^	34.97 ± 0.35 ^gh^
pH 4	52.24 ± 0.63 ^bc^	14.03 ± 0.17 ^c^	12.29 ± 0.27 ^bcd^	39.56 ± 0.43 ^f^
pH 5	53.83 ± 1.93 ^b^	8.37 ± 0.18 ^d^	12.42 ± 0.34 ^bcd^	36.55 ± 1.65 ^g^
pH 6	42.07 ± 0.84 ^ef^	8.09 ± 0.10 ^d^	9.50 ± 0.25 ^f^	47.09 ± 0.76 ^cd^
pH 7	40.48 ± 1.82 ^f^	8.53 ± 0.23 ^d^	11.17 ± 0.72 ^de^	49.00 ± 1.62 ^c^
pH 8	46.65 ± 2.93 ^d^	8.74 ± 0.31 ^d^	12.80 ± 1.73 ^abc^	43.53 ± 2.25 ^e^
pH 9	45.68 ± 4.09 ^d^	3.78 ± 0.14 ^e^	10.22 ± 1.31 ^ef^	43.21 ± 3.70 ^e^
pH 10	43.70 ± 0.90 ^de^	−0.59 ± 0.75 ^f^	12.85 ± 1.09 ^abc^	45.59 ± 0.59 ^de^
pH 11	32.17 ± 0.42 ^h^	−0.10 ± 0.46 ^f^	12.44 ± 0.09 ^bcd^	56.74 ± 0.43 ^a^
pH 12	36.41 ± 1.33 ^g^	−0.12 ± 0.19 ^f^	13.84 ± 0.70 ^a^	52.89 ± 1.11 ^b^
**SPS CA 2%**				
**pH Value**	** *L* **	** *a* **	** *b* **	**Δ*E***
pH 1	50.14 ± 1.37 ^bc^	21.98 ± 0.39 ^b^	11.37 ± 0.41 ^d^	43.32 ± 0.99 ^bcd^
pH 2	49.62 ± 1.03 ^bc^	24.87 ± 0.84 ^a^	11.16 ± 0.55 ^de^	45.27 ± 0.80 ^b^
pH 3	51.28 ± 0.52 ^ab^	15.87 ± 0.40 ^c^	10.08 ± 0.36 ^ef^	39.60 ± 0.43 ^g^
pH 4	48.85 ± 0.44 ^c^	15.92 ± 0.47 ^c^	11.25 ± 0.40 ^de^	41.85 ± 0.32 ^def^
pH 5	45 ± 0.78 ^d^	11.26 ± 0.27 ^d^	9.07 ± 0.36 ^f^	44.13 ± 0.69 ^bc^
pH 6	52.62 ± 0.48 ^a^	8.88 ± 0.14 ^e^	12.70 ± 0.24 ^c^	36.10 ± 0.44 ^h^
pH 7	48.66 ± 0.92 ^c^	8.37 ± 0.07 ^e^	12.12 ± 0.15 ^cd^	39.88 ± 0.91 ^fg^
pH 8	48.36 ± 0.62 ^c^	8.25 ± 0.33 ^e^	11.71 ± 0.24 ^cd^	40.15 ± 0.64 ^efg^
pH 9	36.75 ± 1.33 ^e^	3.41 ± 0.59 ^f^	6.83 ± 1.17 ^g^	51.17 ± 1.26 ^a^
pH 10	44.33 ± 2.05 ^d^	−0.26 ± 0.10 ^h^	12.08 ± 0.41 ^cd^	43.44 ± 2.05 ^bcd^
pH 11	37.19 ± 2.33 ^e^	3.05 ± 0.20 ^fg^	22.00 ± 1.79 ^b^	50.56 ± 2.34 ^a^
pH 12	45.50 ± 1.59 ^d^	2.61 ± 0.12 ^g^	24.43 ± 0.86 ^a^	42.17 ± 1.55 ^cde^

Different superscript lowercase letters within the same row indicate significant differences by the Fisher’s test (*p* < 0.05).

**Table 7 foods-10-02005-t007:** Color response values for the SPS/SPP CA 1%, SPS/SPP CA 1.5%, and SPS/SPP CA 2% films at pH 1–10.

**SPS/SPS CA 1%**				
**pH Value**	** *L* **	** *a* **	** *b* **	**Δ*E***
pH 1	29.75 ± 0.50 ^c^	46.75 ± 0.56 ^a^	43.56 ± 1.65 ^bc^	85.26 ± 0.86 ^d^
pH 2	20.87 ± 0.79 ^e^	35.05 ± 1.52 ^d^	44.9 ± 0.97 ^b^	86.89 ± 0.52 ^cd^
pH 3	32.31 ± 1.44 ^b^	43.64 ± 1.64 ^b^	37.72 ± 0.49 ^d^	79.03 ± 0.38 ^f^
pH 4	38.64 ± 1.11 ^a^	35.79 ± 1.02 ^cd^	21.46 ± 0.59 ^h^	63.72 ± 0.35 ^i^
pH 5	33.98 ± 0.21 ^b^	29.02 ± 0.93 ^e^	26.85 ± 0.66 ^g^	65.96 ± 0.80 ^h^
pH 6	28.73 ± 0.67 ^c^	29.86 ± 0.24 ^e^	34.31 ± 0.62 ^e^	73.63 ± 0.91 ^g^
pH 7	33.47 ± 3.08 ^b^	23.86 ± 1.2 ^f^	31.43 ± 2.05 ^f^	66.24 ± 3.88 ^h^
pH 8	25.92 ± 1.48 ^d^	33.88 ± 1.92 ^d^	42.71 ± 1.95 ^bc^	81.48 ± 0.65 ^e^
pH 9	19.92 ± 0.71 ^e^	37.49 ± 2.41 ^c^	43.95 ± 0.16 ^b^	88.17 ± 0.46 ^bc^
pH 10	11.84 ± 1.80 ^g^	6.21 ± 2.10 ^g^	41.21 ± 2.30 ^c^	85.92 ± 0.67 ^d^
pH 11	14.27 ± 1.41 ^f^	3.2 ± 0.12 ^h^	57.42 ± 2.54 ^a^	92.37 ± 0.44 ^a^
pH 12	8.44 ± 1.37 ^h^	2.02 ± 0.28 ^h^	42.36 ± 2.22 ^bc^	89.26 ± 0.21 ^b^
**SPS/SPS CA 1.5%**				
**pH Value**	** *L* **	** *a* **	** *b* **	**Δ*E***
pH 1	30.02 ± 0.73 ^d^	46.65 ± 0.79 ^a^	47.70 ± 0.10 ^bc^	87.12 ± 0.04 ^b^
pH 2	22.79 ± 0.46 ^g^	42.90 ± 0.25 ^b^	53.39 ± 0.47 ^a^	93.28 ± 0.53 ^a^
pH 3	26.24 ± 0.46 ^f^	41.78 ± 0.14 ^c^	46.35 ± 0.86 ^c^	86.57 ± 0.84 ^b^
pH 4	30.34 ± 0.26 ^cd^	37.6 ± 0.18 ^d^	34.93 ± 0.16 ^e^	76.08 ± 0.15 ^cd^
pH 5	32.58 ± 0.61 ^b^	28.08 ± 0.33 ^f^	29.27 ± 0.08 ^f^	67.66 ± 0.31 ^e^
pH 6	28.1 ± 1.26 ^e^	30.75 ± 0.29 ^e^	35.87 ± 0.23 ^e^	75.20 ± 0.98 ^d^
pH 7	35.10 ± 0.42 ^a^	20.66 ± 0.19 ^h^	30.28 ± 0.16 ^f^	63.30 ± 0.22 ^f^
pH 8	31.57 ± 1.77 ^bc^	26.2 ± 1.35 ^g^	28.64 ± 4.36 ^f^	67.54 ± 3.76 ^e^
pH 9	31.62 ± 0.54 ^bc^	17.38 ± 0.33 ^i^	21.91 ± 1.25 ^h^	62.11 ± 0.81 ^f^
pH 10	19.84 ± 0.14 ^h^	−2 ± 0.04 ^l^	38.69 ± 0.17 ^d^	77.47 ± 0.09 ^c^
pH 11	16.39 ± 0.82 ^i^	2.17 ± 0.03 ^k^	48.90 ± 0.65 ^b^	85.67 ± 0.41 ^b^
pH 12	30.81 ± 0.83 ^cd^	5.49 ± 0.53 ^j^	24.49 ± 0.38 ^g^	61.64 ± 0.83 ^f^
**SPS/SPS CA 2%**				
**pH Value**	** *L* **	** *a* **	** *b* **	**Δ*E***
pH 1	28.56 ± 0.25 ^b^	45.79 ± 0.30 ^a^	48.54 ± 0.10 ^a^	88.10 ± 0.09 ^a^
pH 2	31.45 ± 0.86 ^a^	46.51 ± 0.18 ^a^	46.03 ± 1.50 ^b^	85.23 ± 1.31 ^b^
pH 3	29.23 ± 1.08 ^b^	42.17 ± 1.10 ^b^	42.97 ± 0.22 ^d^	82.93 ± 0.27 ^c^
pH 4	26.17 ± 0.19 ^c^	38.02 ± 0.08 ^c^	41.96 ± 0.03 ^d^	82.67 ± 0.10 ^cd^
pH 5	29.17 ± 1.15 ^b^	28.34 ± 0.46 ^e^	30.52 ± 0.23 ^g^	71.07 ± 0.86 ^f^
pH 6	26.23 ± 0.30 ^c^	28.77 ± 0.56 ^e^	37.06 ± 0.65 ^f^	76.47 ± 0.23 ^e^
pH 7	22.34 ± 1.40 ^d^	33.32 ± 1.76 ^d^	37.17 ± 0.22 ^f^	81.42 ± 1.73 ^d^
pH 8	20.97 ± 0.72 ^e^	37.64 ± 2.79 ^c^	46.53 ± 0.48 ^b^	88.69 ± 1.33 ^a^
pH 9	20.81 ± 0.59 ^e^	39.26 ± 0.16 ^c^	46.26 ± 0.12 ^b^	89.35 ± 0.36 ^a^
pH 10	17.63 ± 0.84 ^f^	−4.64 ± 0.07 ^h^	44.22 ± 0.65 ^c^	82.26 ± 0.43 ^cd^
pH 11	14.42 ± 0.35 ^g^	3.48 ± 0.21 ^g^	38.92 ± 0.80 ^e^	82.39 ± 0.59 ^cd^
pH 12	26.49 ± 0.79 ^c^	12.23 ± 0.77 ^f^	26.26 ± 0.71 ^h^	67.14 ± 0.50 ^g^

Different superscript lowercase letters within the same row indicate significant differences by the Fisher’s test (*p* < 0.05).

**Table 8 foods-10-02005-t008:** Observations on films displaying chicken sample deterioration.

At Hour	pH of Chicken	Chicken in Contact with				
		No Film	SPS CA 0%	SPS CA 2%	SPS/SPP CA 0%	SPS/SPP CA 2%
0	pH 5	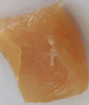	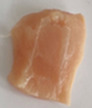	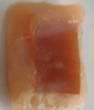	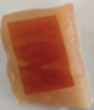	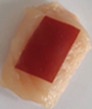
4	pH 6	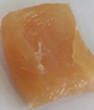	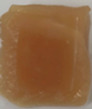	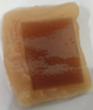	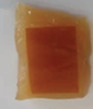	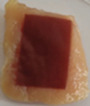
16	pH 7	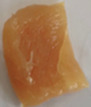	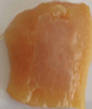	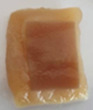	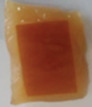	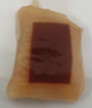
24	pH 8	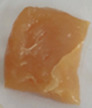	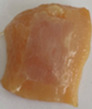	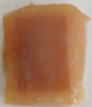	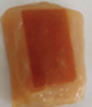	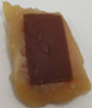

## Data Availability

Not applicable.

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
