# Peer review of "Characterization of Anthocyanin Associated Purple Sweet Potato Starch and Peel-Based pH Indicator Films"

_foods, 2021, doi:10.3390/foods10092005_

Round 1

Reviewer 1 Report

The work entitled: “Characterization of anthocyanin associated purple sweet potato starch and peel-based pH indicator films”, deserves considerable interest, since it takes into consideration sustainable and natural materials. Furthermore, the series of films herein formulated and characterized could offer a promising alternative to reduce the overall environmental pollution.

More in detail, some points of weakness should be improved, as in the following:

  • Generally speaking, bibliography should be extended. The authors are recommended to cite relevant papers in the field:
  • Composites Part A: Applied Science and Manufacturing, Volume 132, 105836 (2020)
  • Composites Science and Technology, 190, 108008 (2020)
  • Water Contact Angle tests should be added, if possible.
  • It would be advisable to insert additional SEM images of the composites, in particular, the sem images of the cross sections should be inserted
  • Scale bar of sem tests should be added
  • It would be advisable to improve the graphics and style of tables and graphs

Reviewer 2 Report

Authors developed smart (pH-sensitive) films based on sweet potato starch and sweet potato peels. The developed films were further enriched with commercial purple sweet potato anthocyanin and some physicochemical, mechanical, thermal, and morphological properties of the films were investigated. In addition, the color response of the indicator films to different pH ranges as well as the migration of anthocyanin from the film to different food simulants was analyzed. Finally, the indicator films were employed to trace the freshness of raw chicken. Overall, the manuscript is well designed. Considering smart packaging and the aim of study monitoring the freshness of food products could be interesting for the industry. Following are some issues leading to the major revision of the manuscript:

  • Line 267: “Direct food contact” this analysis was performed at room temperature for time intervals up to 24 hours. However, the shelf-life of chicken in supermarkets is 6 days at 4°! Your experimental condition is quite far from the real situation. Please add a reference protocol or explanation.
  • Line 319: SEM images should have the same magnification to compare with each other. In addition, the cross-section of the films should be analyzed, this could help authors to discuss also mechanical properties results. What about other anthocyanins concentrations (1 and 1.5%), these concentrations are completely missed in the results? On one side, authors stated that the addition of anthocyanin causes compact morphology (line 3321) and then discussed (line 322) this is due to less compact morphology! The authors should resolve this discrepancy.
  • Line 482: the TS value of films incorporated with CA is less than 5 MP. These values are in the acceptable range for intelligent packaging and the aim of your study monitoring the freshness of food products?
  • Line 581: “ color analysis” the significant color variation in developed films in particular those with high concentrations of CA limited their application for packaging applications since it directly affects consumer acceptance. Are the obtained color values, total color, and whitening index in the acceptable range for packaging? Is there any standard to compare these results?
  • Line 275: please add the number of repeats and replication or if it is different for each analysis, add it to each subsection.
  • Line 212: “28 cm2 can you please control your calculation once again. The transmission area should not be “3.14 x r2” or I am wrong?! 14 x 3.5 x 3.5
  • Line 136: please add speed in rpm.
  • Line 166: what is the size of Petri dishes.
  • Please substitute “gm” with “g” throughout the manuscript. for example, lines 128, 158, 174, 202, etc.
  • Line 175: “A total of 3 mL”
  • Line 179: Please add the L, a, b values of the white standard plate.
  • Line 182: Please add the number of repeats for thickness measurement.
  • Line 188: “slight modification”
  • Line 203: 7 cm diameter.
  • I think sections 2-5 and 2.6.4 considering color measurements can be merge together.
  • Line 236: Please add the number of scans.
  • Line 244: please add the vacuum value?
  • Line 253-357: This can be moved to the discussion.
  • Line 467: “EaB” abbreviations should be defined in their first indication (line 461).
  • Line 588: Figure please correct the caption “visual appearance”
  • Line 637: “Table 4” please control the b* value for pH 1 in this table. Probably, the correct value is “-14.19”.
  • Line 678: “Figure 7” please develop the caption and explain what is the difference between the five images in this figure.
  • Line 682: “Table 8” please substitute the images with pH values.
  • Figures and tables: ALL the abbreviations need to be explained in captions. Same for “CA % (w/v)”

Round 2

Reviewer 2 Report

Please adjust the abstract to 200 words. The graphical abstract is missing.